# Sacituzumab Govitecan in patients with breast cancer brain metastases and recurrent glioblastoma: a phase 0 window-of-opportunity trial

Sacituzumab Govitecan (SG) is an antibody-drug conjugate that has demonstrated efficacy in patients with TROP-2 expressing epithelial cancers. In a xenograft model of intracranial breast cancer, SG inhibited tumor growth and increased mouse survival. We conducted a prospective window-of-opportunity trial (NCT03995706) at the University of Texas Health Science Center at San Antonio to examine the intra-tumoral concentrations and intracranial activity of SG in patients undergoing craniotomy for breast cancer with brain metastases (BCBM) or recurrent glioblastoma (rGBM). We enrolled 25 patients aged ≥18 years diagnosed with BCBM and rGBM to receive a single intravenous dose of SG at 10 mg/kg given one day before resection and continued on days 1 and 8 of 21-day cycles following recovery. The PFS was 8 months and 2 months for BCBM and rGBM cohorts, respectively. The OS was 35.2 months and 9.5 months, respectively. Grade≥3 AE included neutropenia (28%), hypokalemia (8%), seizure (8%), thromboembolic event (8%), urinary tract infection (8%) and muscle weakness of the lower limb (8%). In post-surgical tissue, the median total SN-38 was 249.8 ng/g for BCBM and 104.5 ng/g for rGBM, thus fulfilling the primary endpoint. Biomarker analysis suggests delivery of payload by direct release at target site and that hypoxic changes do not drive indirect release. Secondary endpoint of OS was 35.2 months for the BCBM cohort and 9.5 months for rGBM. Non-planned exploratory endpoint of ORR was 38% for BCBM and 29%, respectively. Exploratory endpoint of Trop-2 expression was observed in 100% of BCBM and 78% of rGBM tumors. In conclusion, SG was found to be well tolerated with adequate penetration into intracranial tumors and promising preliminary activity within the CNS. Trial Registration: Trial (NCT03995706) enrolled at Clinical Trials.gov as Neuro/Sacituzumab Govitecan/Breast Brain Metastasis/Glioblastoma/Ph 0: https://clinicaltrials.gov/study/NCT03995706?cond=NCT03995706.

✉e-mail: brennera@uthscsa.edu

The prognosis for patients with malignant brain tumors is grim. In terms of disease burden, brain tumors of breast origin are frequent, accounting for 15 to 25% of patients with stage IV breast cancer. Nearly half of all women with advanced triple-negative or Her2-positive breast cancer will be diagnosed with brain metastases at some point in their life[1]. Treatment for brain tumors originating from metastatic breast cancer involves surgery, radiotherapy, and systemic therapies. Unfortunately, even these measures are often unsuccessful. On a similar note, glioblastoma multiforme (GBM) is the most common primary brain malignancy in adults representing half of such tumors. In addition to being the most common, GBM is also the most aggressive primary brain tumor, with a median survival of only 20.9 months despite surgery, radiotherapy, chemotherapy and tumor-treating fields[2]. As such, there remains an unmet need for both recurrent GBM (rGBM) and breast cancer with brain metastasis (BCBM). Treatment of both primary and secondary brain tumor is limited by a number of factors, not least of all the selective impedance of the blood-brain barrier, molecular heterogeneity, immunosuppressant tumor microenvironment factors and associated morbidity of eloquent brain involvement. Recently, antibody-drug conjugates (ADC) targeting extracellular receptors such as human epidermal growth factor receptor 2 (HER2) have demonstrated intracranial efficacy with the potential to mitigate, at least partially, many of these challenges[3]. Studies such as KAMILLA, DEBBRAH, and TUXEDO-1 have shown intracranial response rates of 42%, 46%, and 73%, respectively for HER2 targeting ADCs[4-6].

Trop-2, also known as trophoblast cell-surface antigen-2, is a cell surface glycoprotein that is differentially expressed in several epithelial tumors[7,8]. Trop-2 is not expressed by normal brain tissue but 95% of GBM samples show moderate to intense staining as assessed by immunohistochemistry[9]. In addition in an analysis of TCGA data for glioma, Trop-2 expression has been shown to be strongly correlated with proliferation rate, microvessel density. histological grade, and time to death[10].

Sacituzumab Govitecan (SG, IMMU-132, TRODELVY) is an antibody-drug conjugate (ADC) which targets Trop-2 for the selective delivery of SN-38 to tumors. SG consists of a humanized antibody (hRS7) that recognizes Trop-2. This antibody utilizes a pH hydrolysable linker, CL2A, which allows SN-38 to be released at the tumor site[11]. SN-38 is a topoisomerase inhibitor and the active metabolite of irinotecan. For GBM specifically, SN-38 has an IC50 of only $0.00509 \mu M$ which compares favorably with the $0.0363 \mu M$ of irinotecan (https://www.cancerrxgene.org/compound/SN-38/1494/overview/ic50?tissue=GBM). In vitro studies of hRS7 and SG show minimal extracellular release, high payload dissociation and robust antibody-dependent cytotoxicity against Trop-2 positive carcinomas[12-15]. In xenograft models, SG delivers 20 to 136-fold more SN-38 than irinotecan[16,17].

In early-phase trials of pretreated metastatic solid cancers, Trop-2 was found to be highly expressed and SG to be well tolerated[18,19]. The observed half-life for SG in humans was 11 to 14 h, similar to that seen with mice. Among heavily pretreated metastatic TNBC (mTNBC), an overall response rate (ORR) of 33.3% and promising survival signals were observed and this lead to conditional FDA approval[20,21]. This was followed by the confirmatory phase 3 study, ASCENT, where treatment with SG improved OS (11.8 vs 6.9 months PC, HR 0.51) in patients with relapse or refractory mTNBC[22]. Grade 3 or higher treatment-related adverse events (trAE) seen with SG were largely as expected, being neutropenia (51%), leukopenia (10%), diarrhea (10%), anemia (8%), and febrile neutropenia (6%). Similar activity and safety was observed in hormone receptor-positive (HR + ), HER2 negative (HER2-) metastatic breast cancer in TROPiCS-02 and the drug has seen been approved for TNBC and HR + HER2- metastatic breast cancer[23].

Brain metastases remain a clinical dilemma in TNBC. Approximately 50% of all women with advanced TNBC will be diagnosed with brain metastases[24]. The outcome of these patients is quite poor, with a median OS following brain metastasis of only 7.3 months[25]. The available treatment options for mTNBC prior to SG's approval, namely carboplatin and capecitabine, have shown activity within the CNS. However, these have had no impact on OS following diagnosis with brain metastasis, and this stands in stark contrast to other subtypes, such as luminal or HER2 subsets, of patients where more benefit is seen[26]. Given the activity of SG, the novel pH-dependent linker with CNS penetrant payload, and the relatively high frequency of brain metastases in TNBC and the lack of available treatments for this and similar dilemmas in the treatment of rGBM, there is interest in the ability of SG to reach and demonstrate activity against these tumors.

Here, we report the results of a phase 0 window-of-opportunity study of patients with BCBM and rGBM, showing that SG could achieve intratumoral concentrations of SN-38 sufficient for therapeutic benefit in patients with brain metastases from breast cancer and recurrent glioblastoma. The drug was well tolerated in this population with promising clinical signals of efficacy. Additionally, a xenograft model confirmed intracranial activity in mice.

## Results
### SG activity in intracranial Xenografts
Twenty SCID/NCr mice were inoculated intracranially with triple-negative breast cancer MDA-MB-468 cells. The control animals showed rapid tumor growth (Fig. 1, Supplementary Fig. 1) and had all died by 45 days (Supplementary Fig. 2). However, animals treated with SG showed decreased tumor burden, and all remained alive through 60 days a statistically significant difference (P < 0.0001).

### Demographics
Demographics (Table 1) and participant flow (Supplementary Fig. 3) are summarized. 29 patients were screened with 4 screen failures. A total of 25 patients were treated with at least 1 dose of study drug. There were 13 patients in the BCBM cohort and 12 in the rGBM cohort. 1 patient from cohort A (#23) was found on surgery to have pathology not consistent with recurrence.

Among study participants, 22 (88%) were white with 10 (40%) identifying as Hispanic or Latino. The average age of participants was 51.8 years old. 16 (64%) were female as the BCBM population was expectedly female predominant. Of the BCBM patients, 7 (54%) patients had tumors that were HR + , 7 (54%) tumors were HER2+ and 3 (23%) tumors were TNBC. The term 'glioblastoma' is used in this paper to refer to IDH-mutant tumors as the initiation of this study predated the WHO reclassification of this entity to the more proper 'astrocytoma, IDH mutant, CNS WHO grade 4'. 3 rGBM patients (25%) had IDH-mutated tumors (#4, 17, 18). Additionally, 1 patient (#15) had a non-canonical G105G mutation (c.315 C > T). This mutation is of undetermined significance but for demographic purposes this patient is grouped as IDH-wildtype[27]. Interestingly, this G105G mutation could not be verified with 2 separate comprehensive molecular profiling tests but these did reveal a BRAF V600E mutation. Among tumors with MGMT promoter methylation status, 9 (75%) were unmethylated. Two tumors had unknown MGMT methylation status. There was one patient (#6) with WHO CNS grade 2 glioma that was determined to be molecular IDH-wild-type GBM prior to surgery and then found at the time of surgery to be grade 4. Another patient (#19) had grade 2 glioma with no detectable IDH mutation on molecular profiling. All other GBM patient's had tumors which were WHO grade 4 prior to surgery. Additional demographic considerations are summarized in Supplementary Note 1.

### Clinical safety
In this study, Sacituzumab Govitecan was found to be well tolerated. Most AE were grade 1 or 2 (Supplementary Table 1). Among the most common were fatigue (60%), diarrhea (52%), alopecia (44%), headache (40%), neutropenia (40%), and nausea (36%). Grade≥3 AE included neutropenia (28%), hypokalemia (8%), seizure (8%), thromboembolic

Control (Saline/twice weekly) SG (25 mg/kg/twice weekly)

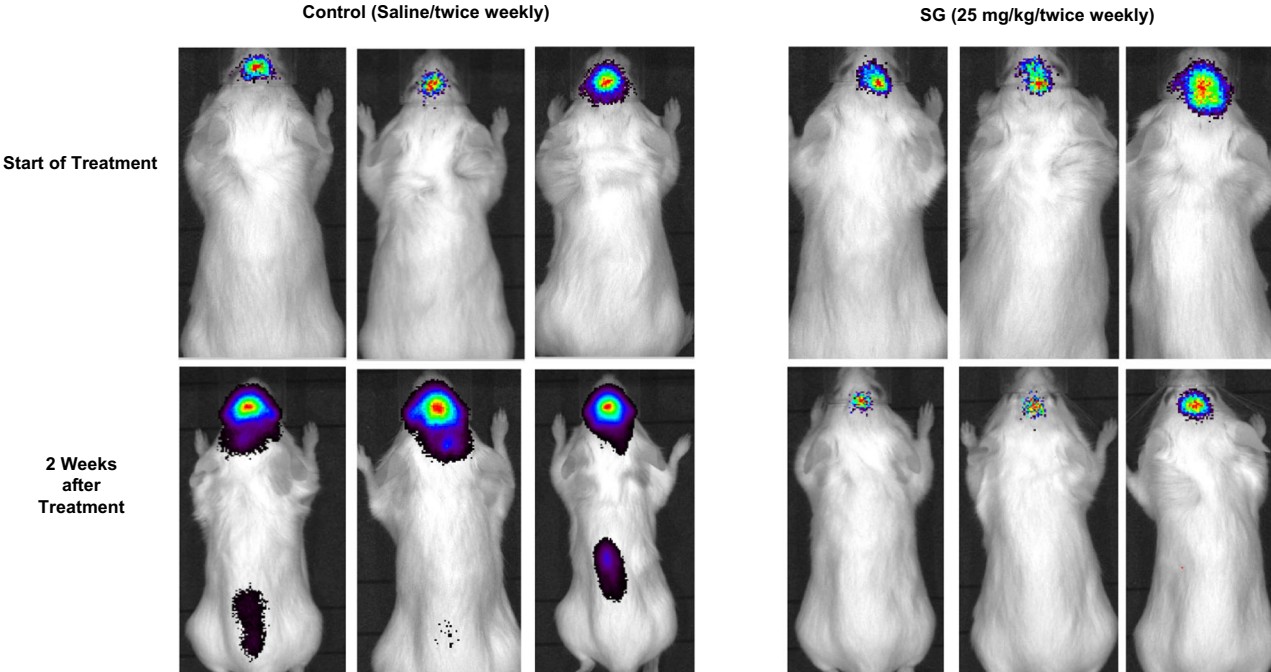

**Fig. 1 | Sacituzumab Govitecan (SG) inhibits tumor growth in a xenograft model of intracranial breast cancer.** Representative bioluminescent images of control and sacituzumab govitecan treated (25 mg/kg/twice weekly) mice at the start of treatment (top panels) and 2 weeks after treatment (bottom panels).

event (8%), urinary tract infection (8%) and muscle weakness of the lower limb (8%). There was one instance of grade 4 neutropenia. One patient in the rGBM cohort had grade 5 pneumonia and this was assessed to be unrelated to the study drug.

### Table 1 | Patient demographics

| Characteristics | No. (% evaluable) | BCBM (Arm A) | rGBM (Arm B) |
|---|---|---|---|
| Number | 25 | 13 | 12 |
| Mean Age (Range) | 51.8 ± 12.3 (33-77) | 48.5 ± 12.2 (33-70) | 55.2 ± 11.9 (38-77) |
| Gender | | | |
| *Male* | 9 (36) | 0 (0) | 9 (75) |
| *Female* | 16 (64) | 13 (100) | 3 (25) |
| Race | | | |
| *White* | 22 (88) | 12 (93) | 10 (83) |
| *Black/African American* | 1 (4) | 1 (7) | 0 (0) |
| *Unknown* | 2 (8) | 0 (0) | 2 (17) |
| Ethnicity | | | |
| *Non-Spanish* | 12 (48) | 5 (38) | 7 (58) |
| *Spanish/Hispanic/ Latino, NOS* | 10 (40) | 8 (62) | 2 (17) |
| *Unknown/Non-disclosed* | 3 (12) | 0 (0) | 3 (25) |
| Histological Features (BCBM) | | | |
| *Hormone receptor positive* | | 7 (54) | |
| *HER2 positive* | | 7 (54) | |
| *Triple negative* | | 3 (23) | |
| Histological Features (rGBM) | | | |
| *IDH wild-type* | | | 9 (75) |
| *MGMT promoter unmethylated* | | | 9 (75) |

Table of demographics showing age, gender, race, ethnicity, and histological features of patients and their respective tumors.

### Clinical efficacy

The observed PFS was 8 months (range 2–26.5 months) for the BCBM cohort (Supplementary Fig. 4). The PFS was 2 months (0.5–13.2 months) for rGBM cohort (Supplementary Fig. 5). The OS for the BCBM cohort (Fig. 2) was 35.2 months (2.7–37 months). The equivalent OS for the rGBM cohort (Fig. 3) was 9.5 months (1–28 months). ORR was a non-planned exploratory analysis. The intracranial ORR (iORR) for the BCBM cohort (Fig. 4, Supplementary Fig. 6, Supplementary Fig. 7) was 50% (37%iSD, 25% iPR, 25% iCR). The extracranial response is also summarized (Supplementary Fig. 8). The ORR for the rGBM cohort (Fig. 5, Supplementary Fig. 9, Supplementary Fig. 10) was 28% (28% SD, 28% PR, 0% CR).

### Quantification of SN-38 levels in Serum, Tissue and CSF

SN-38 levels, the primary outcome of interest, were quantified in 24 matching samples of tissue and serum, as well as 4 corresponding patients who had CSF available (Supplementary Table 2). One patient (patient 16, rGBM) had insufficient samples for SN-38 (serum and tissue) analysis. Total SN-38 levels in the BCBM ranged from 1266.8 to 5659.6 ng/ml (median 2462.4 ng/ml, IQR 2483.2) in serum and 86.5 to 652 ng/g (median 197.3 ng/g, IQR 230.1) in tissue. Assuming BCBM tissue density of 1.04 g/mL (brain), this corresponds with a molarity of 6.27 μM and 0.0523 μM for serum and tissue, respectively[28]. For reference, the minimum IC50 for SN-38 (MW 392.4 g/mol) in invasive breast cancer cell lines is 0.00517 μM[29,30]. The patient in the BCBM whose pathology was determined not to be recurrence had a total SN-38 level in tissue of 21.2 ng/g. Correspondingly in GBM patients, total SN-38 levels ranged from 115 to 5363.1 ng/ml (median 2465.7 ng/ml, IQR 1992.9) in serum, and 8.6 to 259.1 ng/g (median 104.5 ng/g, IQR 182.7) in tumor tissue. These findings correspond with a molarity of 6.28 μM and 0.28 μM for serum and tissue, respectively. For reference, the minimum IC50 for SN-38 (MW 392.4 g/mol) in GBM cell lines is 0.00468 (μM)[29,30]. Three patient CSF samples were collected in the BCBM cohort, and total SN-38 ranged from 5 to 26.5 ng/ml (median 9.4 ng/ml). There were one CSF sample collected in the GBM cohort also, and total SN-38 was 5.1 ng/ml.

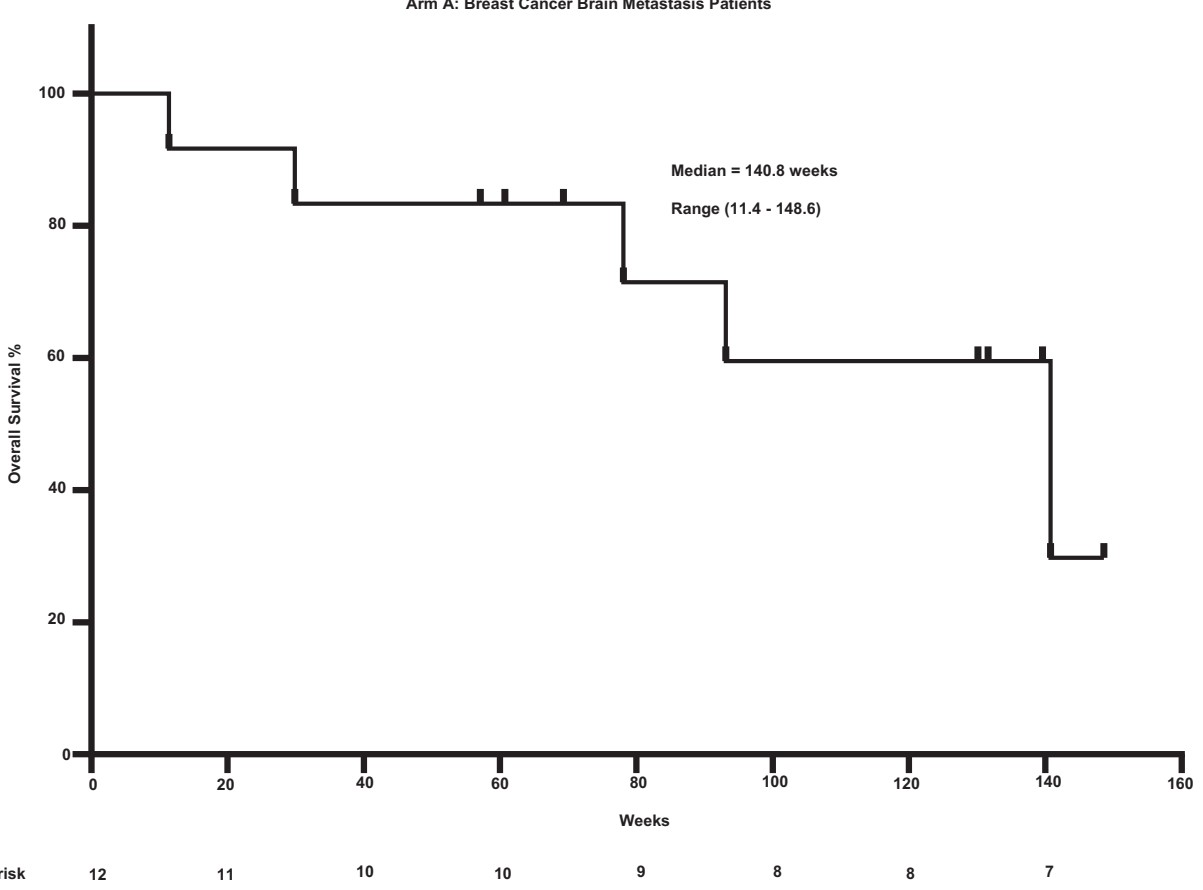

**Fig. 2 | Overall survival for patients having breast cancer with brain metastasis after treatment with sacituzumab govitecan.** Kaplan-Meier plot showing overall survival for patients ($n = 12$).

### Trop-2, CAIX and γH2AX expression in patient tumor tissue

Patient tumor samples were quantified as to the expression of Trop-2 and carbonic anhydrase IX (CAIX) by immunohistochemistry with semiautomated image analysis and scoring. There were 12 BCBM and 12 rGBM samples available for this (Supplementary Table 2). Two patients (#12, BCBM; #17, rGBM) had insufficient tissue for IHC (Trop-2, γH2AX, CAIX). Additionally, three rGBM patients whose samples showed radiation injury with reactive cells (#4, #18, #22) and were thus not sufficient for consideration of Trop-2 or CAIX expression. One patient (#16) had no sample for SN38 quantification.

All of the BCBM samples were given an H-score of 3 + , meaning that >75% of tumor cells staining positive for Trop-2. The intensity of the Trop-2 stain was either strong (9/11) or medium (2/11). Trop-2 staining was observed mostly on the membrane with a few samples also having cytoplasmic staining (Supplementary Fig. 11). The 9 sufficient rGBM samples had an H-score of 3+ (1/9), 2+ (2/9), 1+ (4/9) and 0 (2/9) for the overall number of tumor cells staining positive for Trop-2. For the intensity of staining, the majority of the samples scored either weak (6/9) or medium (2/9). Trop-2 staining distribution in the GBM samples was mostly in the cytoplasm (Supplementary Fig. 12). The Pearson r for the percent of SN-38 tissue/serum ratio vs Trop-2 expression in the BCBM cohort (Supplementary Fig. 13) was 0.42 ($r^2 = 0.018$, 95% CI = −0.23 to 0.81, $p = 0.18$). For the rGBM cohort (Supplementary Fig. 14), the Pearson r for the percent of SN-38 tissue/serum ratio vs Trop-2 expression was 0.85 ($r^2 = 0.73$, 95% CI = 0.29 to 0.97, $p = 0.013$).

γH2AX expression was also quantified.

Prespecified exploratory analysis was undertaken to better understand the mechanism of action for SG. For the BCBM cohort the percent positive nuclei were 54.4 for the BCBM cohort and 73.6 for the rGBM cohort. The Pearson r for the percent of SN-38 tissue/serum ratio vs γH2AX expression in the BCBM cohort (Supplementary Fig. 15) was 0.25 ($r^2 = 0.065$, 95% CI = −0.41 to 0.74, $p = 0.45$). For the rGBM cohort (Supplementary Fig. 16), the Pearson r for the percent of SN-38 tissue/serum ratio vs γH2AX expression was 0.002 ($r^2 = 3.7e^{-6}$, 95% CI = −0.75 to 0.75, $p = 0.99$). Regarding CAIX expression, in the BCBM cohort, 5/11 (45%) samples showed high (>10% positive tumor cells) expression of CAIX staining. In the GBM cohort, only 3/8 (38%) had high CAIX expression. The percent of SN-38 tissue/serum vs CAIX was also dichotomized for the BCBM (Supplementary Fig. 17) and rGBM cohorts (Supplementary Fig. 18).

### Discussion

Preclinical evaluation in an intracranial TNBC murine model, showed that SN-38 has antitumoral activity against these brain tumors with resulting survival benefit and supporting the overall hypothesis.

In this study, preoperative Sacituzumab Govitecan was determined to accumulate to considerable amounts in the intracranial tumor tissue of both brain metastasis from breast cancer and recurrent glioblastoma. There are several completed and ongoing clinical trials of ADC in brain metastasis, predominantly breast cancer[31]. However, few have sought to quantify intratumor drug concentrations. Unsurprisingly, for both BCBM and rGBM, tissue concentrations of total SN-38 were less than those observed in serum. Despite this, the levels achieved were noted to well exceed, by an order of magnitude, the established IC50 of representative cancer cells in vitro. Thus, the study met its prespecified primary endpoint. In humans (IMMU-132-01,

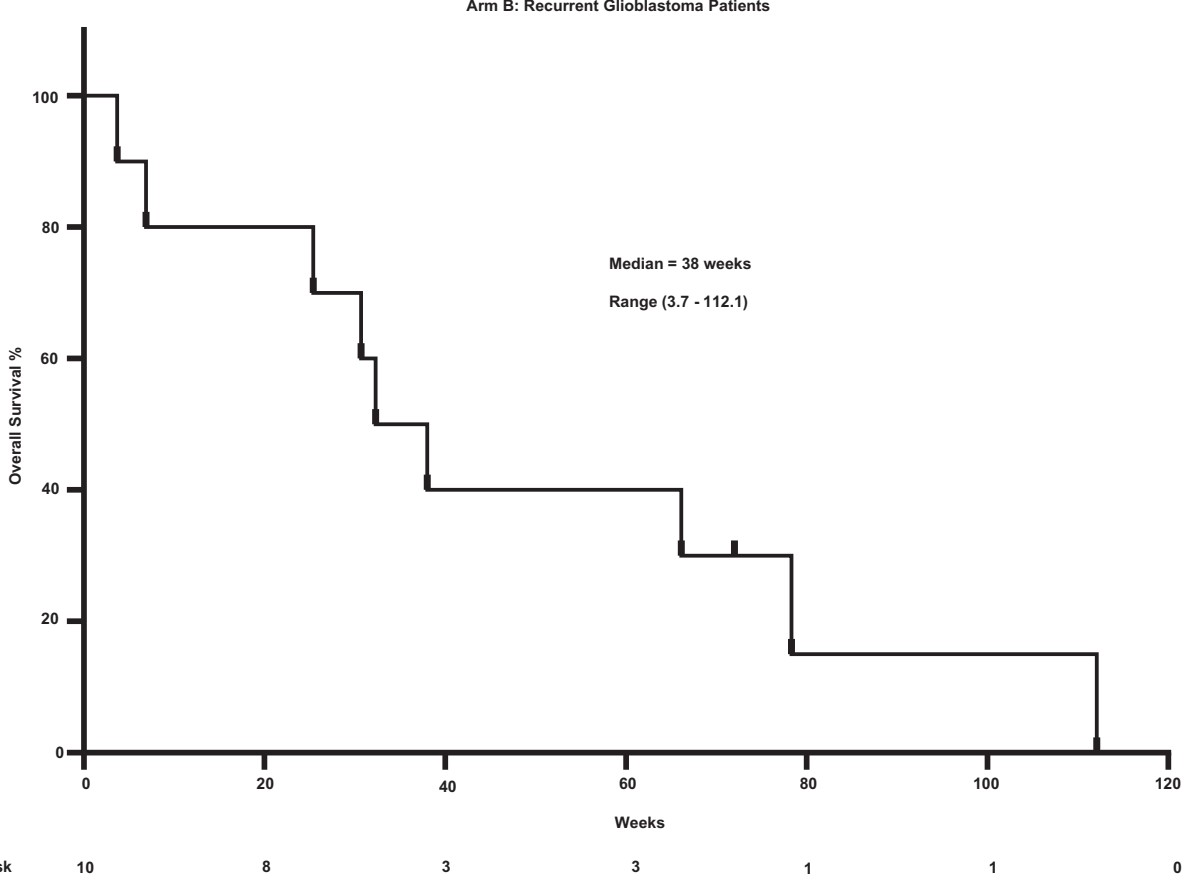

**Fig. 3 | Overall survival for patients with recurrent glioblastoma after treatment with sacituzumab govitecan.** Kaplan-Meier plot showing overall survival for patients (*n* = 10).

IMMU-132-05), the Cmax for total SN-38 has been reported to be 227,000-240,000 ng/mL which was more than we observed[32]. However, it should be noted that the Tmax for SG is 3.3 h. Our study was not a dedicated pharmacokinetic study and only one timepoint was obtained. Accordingly, patients received a preoperative dose with tissue and serum collection the next day. Therefore, it is likely that the observed total SN-38 concentrations were during the elimination phase of drug exposure.

For patients with BCBM, we observed a median PFS and OS of 8 and 35.2 months, respectively. Patients in the ASCENT trial of metastatic TNBC were eligible with brain metastasis provided they had stable CNS disease for at least 4 weeks prior to treatment. This was a small subset of patient (12%), and therefore limited predictive value, but the median OS was 6.8 months (S7[33])[34]. For patients with HER2 + BCBM, the median OS is 30 to 38 months depending upon whether the CNS metastasis are observed at or after diagnosis[35]. For HR + BCBM, median OS is 12.5 months[36]. A limitation of this window-of-opportunity study is that the sample size and broad inclusion of breast cancer subtypes places reservations on drawing strong conclusions about survival for any one subtype. Given the promising activity, further characterization of CNS activity for BCBM is being evaluated in SWOG S2007.

In the rGBM cohort, the average PFS and OS observed was 2 and 9.5 months, respectively. For comparison, the average OS observed after treatment with lomustine and bevacizumab in EORTC 26101 was 9.1 months[37]. This difference is potentially noteworthy given the high proportion of MGMT unmethylated patients (77%). MGMT unmethylated patients, even when newly diagnosed and despite initial therapy with chemoradiation, have a median OS from initial treatment of only

12.7 months[38]. MGMT unmethylated patients treated with TTFields show an OS of 16.9 months when used in the newly diagnosed setting and only 6.5 months in the recurrent[39,40]. Care should be taken in generalizing this OS too broadly as our population had 3 IDH-mutant gliomas, and one who had a silent IDH mutation, BRAF and three-year survival, suggesting the possibility of alternative pathology such as epithelioid GBM or PXA.

In the study population, SG was found to be well tolerated. The most common observed AE were fatigue, diarrhea and alopecia, all of which were commonly observed among patients on the ASCENT trial. Interestingly, the observed rates of neutropenia were less than previously described[41]. And this may be attributable to presurgical growth factor support, which was mandated in the protocol.

For this study, total SN-38 levels in resected tissue samples showed good delivery of the payload to tumor. Although it is only one data point, it should be noted that the one patient whose surgical tissue was not consistent with recurrence (patient 23) had the lowest level of detected SN-38, consistent with the hypothesis that SG effectively delivers chemotherapy to the target while sparing normal parenchyma. CSF levels were overall low, a likely consequence of minimal Trop-2 expression in CSF space, although low numbers preclude a definitive statement. Correlative biomarkers were examined to better elucidate the mechanism of action of SG in brain tumors. More specifically, the goal was to provide evidence for or against two competing hypotheses. Either that, SG would deliver SN-38 directly at the tumor cell by binding and internalizing the antibody complex with resultant DNA damage or, alternatively, that the pH-dependent linker would release SN-38 in the presence of a highly-acid tumor microenvironment. To understand this better, we sought to correlate markers of

Intracranial Best Response – Arm A: Breast Cancer Brain Metastasis Patients

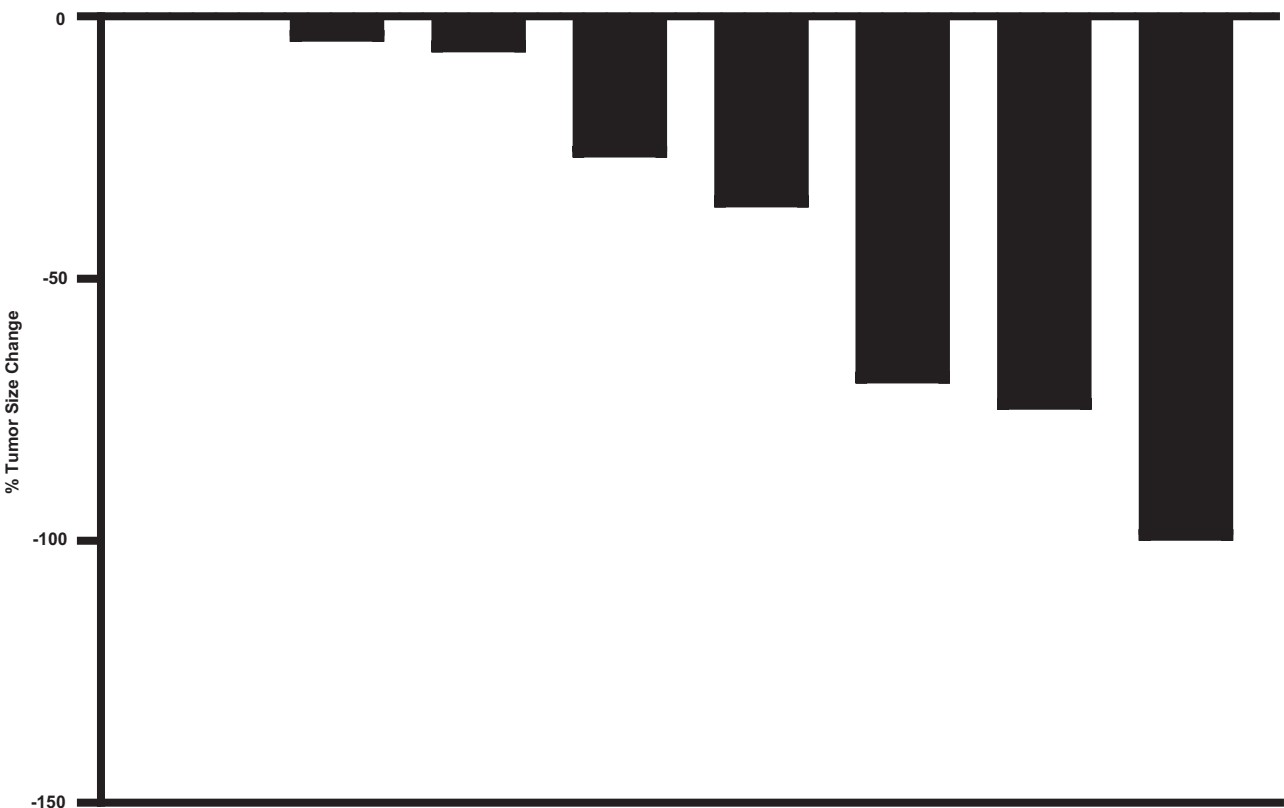

**Fig. 4 | Best intracranial response for patients having breast cancer with brain metastasis after treatment with sacituzumab govitecan.** Waterfall plot showing best intracranial response in tumor size change from baseline for each patient ($n = 8$).

antigen expression (Trop-2), DNA damage (γ-H2AX) and intratumoral hypoxia (CAIX). Correlation between tumor expression of Trop-2 and DNA damage stemming from the arrest of topoisomerase I on DNA, as measured by the DNA damage marker γ-H2AX, would indicate payload release. Conversely, correlation between free SN-38 and hypoxia, as assessed by carbonic anhydrase IX (CAIX) levels, would support the second premise of pH-dependent release of SN-38 within the acidic microenvironment. These biomarkers have previously been shown by ourselves and others to be good surrogates to their respective phenomena[42]. As expected, Trop-2 expression correlated with % SN-38 tissue/serum ratio for both BCBM and rGBM cohorts indicating deliver of payload by the antibody. γ-H2AX did not correlate with % SN-38 tissue/serum ratio. However, a lack of correlation could be explained by prior alkylating therapy causing high levels of preexisting DNA damage or a failure of the SN-38 to sufficiently impact DNA in the 1-day interval between dosing and surgery. CAIX did not correlate with % SN-38 tissue/serum ratio, suggesting hypoxia does not appreciably drive indirect SN-38 release. Indeed, recent evidence from other groups has also called this pH-dependent mechanism into question but more data is needed[43].

Potential sources of bias in this window-of-opportunity study include selection bias due to lack of randomization, ascertainment bias due to lack of binding and attrition bias as some patients were withdrawn and lost to follow-up. Therefore, further study and confirmation is needed.

In conclusion, in this phase 0 study of patients with brain metastasis from breast cancer and recurrent glioblastoma, pre-surgical Sacituzumab Govitecan given preoperatively and as adjuvant therapy to surgery, was determined to be well tolerated with robust efficacy signals. Most importantly, encouraging intratumoral concentrations were achieved. This data supports ongoing investigation in a phase 2 clinical trial (NCT04559230) of this drug in recurrent glioblastoma.

## Methods
### Ethics
The study was approved by the UTHSA IRB at the University of Texas Health Science Center at San Antonio. No restrictions on maximal tumor size or burden were made. The study was conducted in accordance with the Department of Health and Human Services, the Declaration of Helsinki and all relevant ethical regulations. Informed consent was obtained from each patient prior to commencement of study activities. All animal experiments were performed using a UTHSCA IACUC-approved protocol and according to all relevant ethical regulations.

### Xenograft model
Twenty SCID/NCr immunocompromised mice were inoculated intracranially with a $2 \times 10^6$ count of a triple negative breast cancer cell line (MDA-MB-468-GFP-Luc). Cell line was sourced from ATCC HTB-134 and tested for mycoplasma but not authenticated. MDA-MB-468 does not appear in misidentified cell lines in the ICLAC register (version 13). Mice were purchased from Charles River Laboratories and housed in a dedicated pathogen-free cages with a cycle of 12 h light and 12 dark and 22 °C and 50% humidity. All mice were 6 weeks at time of inoculation. Only female mice were used as this was a model of breast cancer metastasis and men account for less than 1 percent of breast cancer incidents in humans. Two weeks

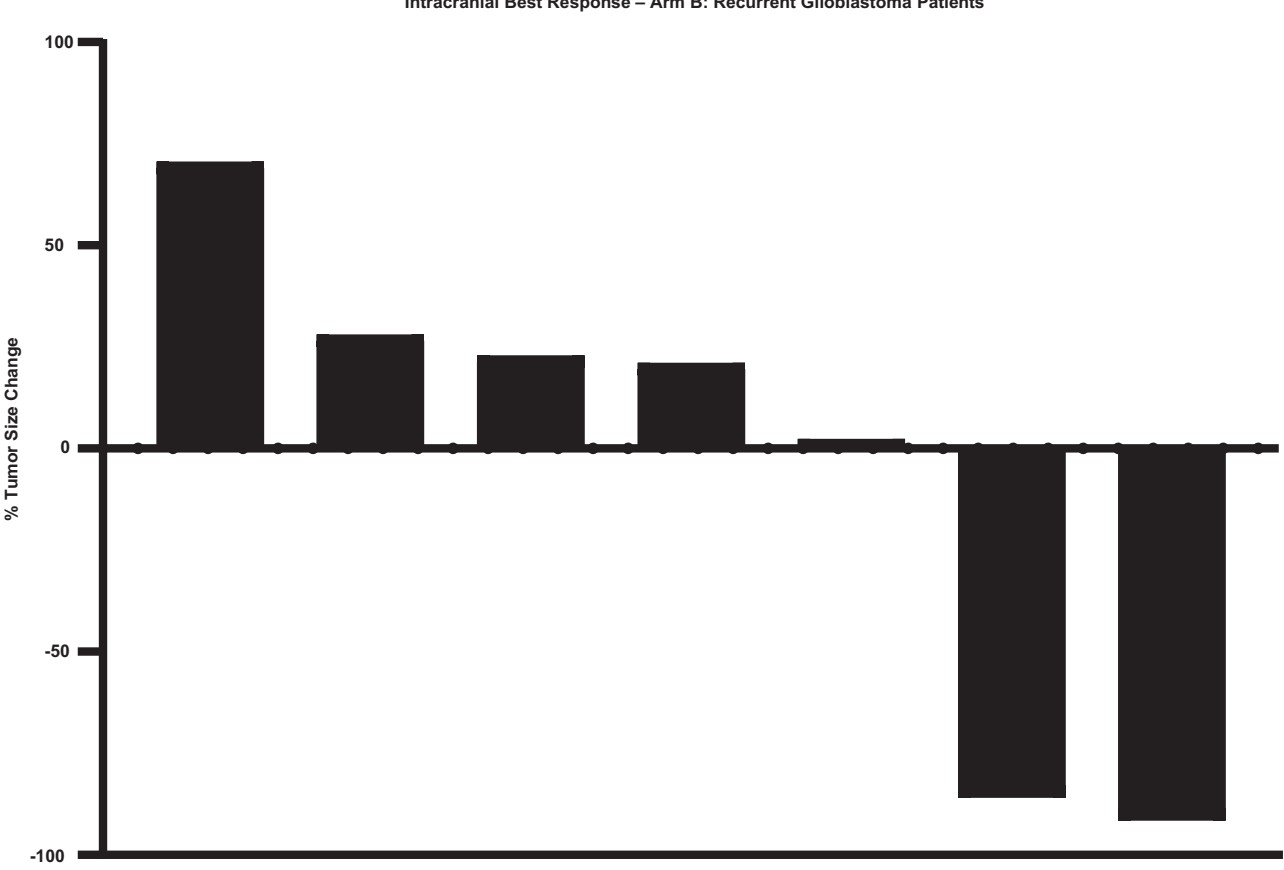

**Fig. 5 | Best intracranial response for patients with recurrent glioblastoma after treatment with sacituzumab govitecan.** Waterfall plot showing best intracranial response in tumor size change from baseline for each patient ($n = 7$).

after intracranial inoculation, tumor volume was measured by a Xenogen in vivo imaging system and animals were then randomized into groups of 10 and treated with 25 mg/kg of SG or vehicle (saline) twice a week for 3 weeks.

## Study design

This phase 0, prospective, single-center, non-randomized, window-of-opportunity study (NCT03995706) examined SG in patients undergoing craniotomy for breast cancer with brain metastases or recurrent glioblastoma. Patient enrollment took place from 8/16/19 to 10/22/20 at the Mays Cancer Center. Mays Cancer Center is the only NCI-Designated cancer center in South Texas and part of the University of Texas Health Science Center at San Antonio (UTHSCSA). Data collection continued through 9/2023. Length of follow-up was 1 year. There was no blinding due to study design and risks of surgery. The primary endpoint was to determine intracranial concentrations of SN-38 as well as those of serum and cerebrospinal fluid (CSF). Secondary endpoints were measuring the PFS and OS for these patients from first post-surgical treatment as well as to assess safety in this population. While not statistically powered, the exploratory IHC analysis of Trop-2, CAIX and γH2AX expression and correlation was pre-specified. Up to 30 subjects could be enrolled. Full and most recent protocol is available in the Supplementary Files under Supplementary Note 2. Trial was enrolled at Clinical Trials.gov as Neuro/Sacituzumab Govitecan/Breast Brain Metastasis/Glioblastoma/Ph 0: https://clinicaltrials.gov/study/NCT03995706?cond=NCT03995706. Study preregistered 6/17/19. Primary study activities conducted from 7/17/19 to 6/3/22. The pre-registered trial protocol is also available in the Supplementary Files under Supplementary Note 3.

## Treatment

Patients received a single dose of SG at 10 mg/kg intravenously (IV) the day prior to craniotomy (Surg D-1). A serum half-life of 11-14 h was observed in the phase ½ study[19]. Therefore, a pre-surgical interval of roughly 24 h was deemed appropriate to measure intracranial penetration while accounting for operating room logistics. Tumor specimens and CSF were collected intra-operatively alongside contemporaneous blood serum. Patients resumed SG 10 mg/kg IV on day 1 (D1) and D8 of 21-day cycle with cycle 1 (C1) beginning upon recovery. Initial infusion was over 3 h with allowance for shortening to 1-2 h on subsequent infusions if well tolerated. All patients were pre-medicated with acetaminophen, diphenhydramine, and dexamethasone. Patients continued treatment until significant trAE, study withdrawal or progressive disease (PD). Growth factor support (Neulasta) was given on Surg D-1 and optionally at D8 of each cycle depending upon absolute neutrophil count (ANC) at the investigator's discretion.

## Eligibility

Only patients with surgical plans, as determined by the neurosurgical and medical oncology teams, for non-emergent craniotomy based on standard of care treatment of their disease were eligible for this study. Cohort A enrollment required histologically or cytologically documented breast cancer of all subtypes (including HR+, HER2 + and TNBC) in addition to known or suspected parenchymal brain metastases. Cohort B required GBM with documented progression by RANO criteria following standard combined modality treatment with radiation and temozolomide. Patients eligible were 18 years or older with the capacity and willingness to sign consent. Additionally eligible

patients had good performance status, life expectancy ≥3 months, recovery from prior trAE grade ≥2, as well as good organ and hematological function. Exclusion criteria excluded patients receiving coumadin anticoagulation, enzyme-inducing anti-epileptic agents, biologic agents (within 21 days of first dose), or prior SG. Moreover, as SG is metabolized by UDP-glucuronosyltransferase, UGT1A1 inhibitors or inducers were not allowed. Lastly, patients with leptomeningeal carcinomatosis were also excluded. Notably, multiple, posterior fossa or bihemispheric metastasis were not excluded from either cohort.

## Safety and efficacy

All patients were screened and monitored throughout the study for AE using CTCAE version 5. Patients were assessed by MRI and CT imaging at screening, during the postsurgical visit and, from cycle four on, every third cycle. Cohort A was assessed for bicompartmental progression simultaneously with Response Assessment in Neuro-Oncology for Brain Metastasis (RANO-BM) for unresected or partially resected CNS disease and Response Evaluation Criteria in Solid Tumors (RECIST), version 1.1 for non-CNS disease. Cohort B was assessed by (RANO 1.0) alone[44]. ORR, PFS, and OS were calculated from C1D1. All patients received routine laboratory testing, physical examinations, vital signs, and assessment of Eastern Cooperative Oncology Group (ECOG) performance status. At screening a 12-lead ECG was also performed. Urinalysis was performed at screening and D1. Pregnancy tests for women of child-bearing potential were performed at screening and D1 of each odd cycle. MRI scans were performed on 3 T MRI scanners (Philips, GE, or Siemens). Although not dictated by the protocol, each session usually consisted of 3D pre- and post-contrast T1 weighted images, FLAIR (fluid-attenuated inversion recovery), and diffusion weighted images. T1 pre-contrast, FLAIR images were acquired before administration of contrast agent.

## Statistical consideration

Given the paucity of available data regarding ADC uptake in human tumors and the probable heterogeneity of this, no formal sample size calculations were performed. Descriptive statistics were used to summarize and compare drug concentrations in serum, cerebral spinal fluid (CSF) and tumor tissue. Drug levels from resected specimens were presented as averages and ranges of determinations, with correction for blood volume using the absorbance of hemoglobin. Tumor concentrations were expressed as tumor-to-serum ratios. For serum studies area-under-the-curve (AUC) concentration was calculated, ending at the time of resection. The number of patients screened, screen failures by reason, number enrolled and completing the study at each stage as well as the number and proportion progression-free at each stage was tabulated. The distribution of time to progression and death was summarized with Kaplan-Meier curves. AE were tabulated. All statistical testing was two-sided with a significance level of 5%. For this paper, results have been rounded to the nearest whole number where appropriate. SAS Version 9.3 for Windows (SAS Institute, Cary, North Carolina) was used throughout. REDCap (multiple versions up to 13.7.31) was used for data collection and monitoring throughout. Microsoft 365 (Version 1904 to 2405) with Word, Excel and PowerPoint were used for analysis and manuscript drafting.

## Immunohistochemistry methods and statistical interpretation

IHC was performed on formalin-fixed, paraffin-embedded whole tissue sections (4 μm) of all specimens using a polyclonal goat anti-Trop-2 antibody (R&D Systems, Catalog: AF650, Clone: not provided, Lot: CIE0319091) at a concentration of 5 μg/mL (Dilution 1 to 40). Immunoreactivity was visualized using DAB substrate and counterstained with hematoxylin (Vector Laboratories Inc., Burlingame, CA). Tissue blocks containing the most representative and well-preserved tumor areas were selected for immunostaining. Tonsil tissue was used as a positive control. Representative sections were subjected to IHC for carbonic anhydrase IX (CAIX) using a kit (WILEX Oncogene Science, CA IX IHC Kit, Catalog: 06490035, Clone: not provided, Lot: 776391 A) for quantification of hypoxia as previously described[45]. For γH2AX analysis, formalin-fixed and paraffin-embedded patient tumor samples were deparaffinized and stained with an antibody against Phospho-Histone γH2AX ([Ser139][20E3], Cell Signaling, Catalog: 9718 S, Clone: not provided, Lot: 21, Dilution: 1 to 100). Slides were visualized using DAB and counterstained with hematoxylin, and then imaged using Leica Aperio VERSA200 at 20x magnification.

For the statistical interpretation of Trop-2 expression by IHC, two means of analysis were performed. First, a semiquantitative analysis by a pathologist utilizing the H-score of 3+, 2+, 1+ or 0, and intensity of strong, medium or weak. Secondly, semiautomated image analysis and semiquantitative scoring were performed using Aperio eSlide Manager. The score is obtained by the formula: 3 x percentage of strongly staining nuclei + 2 x percentage of moderately staining nuclei + percentage of weakly staining nuclei, thus giving a range of 0 to 300. The hypoxic fraction was calculated as the ratio of area of the region of interest, identified with morphometric analysis, to the total area of analysis. The analyses between IHC and drug concentration correlation was calculated by linear regression.

## Pharmacodynamic methods

Serum and tissue samples were processed using established methodology[46–49]. Briefly, tissue samples were harvested, washed, snap frozen in liquid nitrogen, pulverized using a Precellys 24 beads-based tissue homogenizer equipped with Cryolys cooling system (Bertin Technologies, Montignyle-Bretonneux, France) and extracted using a precipitating reagent composed of methanol, ethylene glycol and zinc sulfate as has been previously reported[50]. The polar fraction was then dried prior to re-dissolving in the appropriate solvent for mass spectrometry analysis[48]. Bradford protein assay was used for protein quantification[51]. Plasma samples were deproteinized using a modified Bigh-Dyer procedure[52]. Quantification of total SN-38 level (free + antibody bound) was performed as previously described by our laboratory[53]. Briefly, free SN-38 and SN-38G concentrations were analyzed using stable isotope dilution ultrahigh-performance liquid chromatography–high resolution mass spectrometry (UHPLC-HRMS). In addition, the amount of antibody-conjugated SN-38 was quantified by acid hydrolyzing separate aliquots of both serum and tissue samples to release SN-38 and the total amount of SN-38 was then measured[16]. The calibration standards were prepared by serial dilution of stock solution of these targeted analytes in blank tissue matrices within ranges from 0.1 to 6000 ng/ml. Internal standards included deuterated forms of SN-38-$d_3$ (Sigma Aldrich) and SN-38G-d3 (Santa Cruz Biotechnology) and spiked in each calibration standard at a concentration of 250 ng/ml[54]. The linearity was determined by calculating a regression line using the method of least squares analysis.

The UHPLC-HRMS/MS analysis was performed on a Vanquish Fles UHPLC system coupled to a hybrid quadrupole-Orbitrap mass spectrometer (Q Exactive, Tehrmo Scientific, Waltham, MA) via an electrospray ionization source. Chromatographic separation of targeted analytes and internal standard was achieved on a Kinetex C18 150 × 2.1 mm (2.6 μm, 100 Å) column (Phenomenex, Torrance, CA). Samples were eluted with a binary solvent system with 0.1% formic acid (A) and 0.1% formic acid in acetonitrile (B) using the following linear gradient separation: buffer B was increased from 3% to 60% in 15 min, washed with 85% B for 5 min and equilibrated for 10 min with 3% buffer B before the next injection. Detection and determination were performed in full MS/AIF mode with positive electrospray ionization mode. The targeted multiplex SIM scans for quantification included SN-38 ($m/z$ 393.146), and SN-38G (568.535)[55,56]. Optimized MS parameters were as follow: spray voltage, 4.0 kV; capillary temperature,

300 °C; sheath gas, 50 (arbitrary units); auxiliary gas, 10 (arbitrary units); microscans, 1; maximum injection time, 200 ms; AGC target, 1e6/5e5; mass resolution, 140,000/70,000 FWHM; m/z range, 150e1000; higher energy collisional dissociation energy; 22 eV. Nitrogen was used as a collision gas. The mass spectrometer was calibrated before analysis using commercial calibration solutions to maintain mass accuracy below 5 ppm. The Xcalibur 2.2 software (Thermo Scientific, Waltham, MA, USA) was used to control the instrument and for data acquisition and processing. Q Exactive 2.2 SP 1 (Thermo Scientific) was used to control the tuning window of mass spectrometer.

## Reporting summary
Further information on research design is available in the Nature Portfolio Reporting Summary linked to this article.

## Data availability
In order to protect potential indirect identifiers while also supporting scientific endeavors, the patient Source Data is not available with this manuscript but the data generated in this study are available upon appropriate request from the corresponding author. Source Data for non-patient studies are available. Requests can be made for preclinical and deidentified patient data. Email is the preferred mode of contact and requests should be made from publication and up until 36 months following publication. Requests for non-commercial analysis should be made by researchers and include sound justification such as for use in meta-analysis. Signed data access agreements may be required. Full protocol is available in the Supplementary Files under Supplementary Note 2. Source data are provided with this paper.

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

## Acknowledgements

Sincere thanks to our patients for their trust and participation in this study. We are grateful to Mohamed I Gadallah from UT Austin, Texas for assistance at various times toward the completion of this work. Equal contribution by HB and WK for basic science and clinical efforts, respectively. A METAvivor Metastatic Translational Research Award (Minneapolis Metsquerade Presented In Memory of Tarah Harvey 2019, AB) was the primary funding mechanism for this study. Additional support provided by an NCI Cancer Center Support Grant (P30CA016672, MDA) and an Early Clinical Investigator Award through CPRIT (RP210164, WK). We thank that sponsor, Gilead, for related funding support and providing drug. Gilead was not involved in study design, data collection, analysis but were allowed to make minor editing suggestions on posters and the manuscript with the authors having final discretion over acceptance or refusal of these suggestions.

## Author contributions

Experimental Design: P.S., A.B., Implementation and Execution: W.K., V.K., K.L., M.M., JF., A.B. Analysis and Interpretation: H.B., W.K., G.S., J.M., A.G., P.G., S.T., R.P., J.C., AL., A.B. Literature Review and Manuscript Drafting: H.B., W.K., P.G., A.B.

## Competing interests

HB, WK, KL, PG, GS, JM, AG, PS, ST, RP, JC, AL, JF and AB have no personal, professional or financial relationships that would constitute a conflict of interest. VK and MC are on the speaker's bureau and VK is a consultant for Gilead. Portions of this work have previously been presented in abstract form at the 2023 Society for NeuroOncology Annual Meeting, San Antonio Breast Cancer Symposium 2022 and 2024, and the European Society for Medical Oncology Virtual Congress 2020. WK and AB have received contracted funding from Gilead for a similar study.

## Ethics and inclusion statement

This research has included local researchers throughout the research process and is locally relevant with 40% of the patients identifying as Hispanic or Latino.

## Additional information

Inclusion & Ethics Statement: This research has included local researchers throughout the research process and is locally relevant with 40% of the patients identifying as Hispanic or Latino.

Henriette U. Balinda[1,11], William J. Kelly [1,11], Virginia G. Kaklamani[1], Kate I. Lathrop[1], Marcela Mazo Canola[1], Pegah Ghamasaee[2], Gangadhara R. Sareddy [1,3], Joel Michalek[4], Andrea R. Gilbert[5], Prathibha Surapaneni[6], Stefano Tiziani [7,8,9,10], Renu Pandey[7,8], Jennifer Chiou[7,8], Alessia Lodi[7,8], John R. Floyd II[2] & Andrew J. Brenner[1] ✉

[1]Mays Cancer Center at UT Health San Antonio, 7979 Wurzbach Road, San Antonio, TX 78229, USA. [2]Department of Neurosurgery, University of Texas Health Science Center at San Antonio, 7703 Floyd Curl Drive, San Antonio, TX 78229, USA. [3]Department of Obstetrics & Gynecology, University of Texas Health Science Center at San Antonio, 7703 Floyd Curl Drive, San Antonio, TX 78229, USA. [4]Department of Population Health Sciences Greehey Children's Cancer Research Institute, University of Texas Health Science Center at San Antonio, 8403 Floyd Curl Drive, San Antonio, TX 78229, USA. [5]Department of Pathology and Laboratory Medicine, University of Texas Health San Antonio, 7703 Floyd Curl Drive, San Antonio, TX 78229, USA. [6]START Center for Cancer Care, 155 E Sonterra Blvd STE. 200, San Antonio, TX 78258, USA. [7]Dell Pediatric Research Institute, Dell Medical School, The University of Texas at Austin, Austin, TX 78723, USA. [8]Department of Nutritional Sciences, The University of Texas at Austin, Austin, TX 78712, USA. [9]Department of Oncology, Dell Medical School, Livestrong Cancer Institutes, The University of Texas at Austin, Austin, TX 78723, USA. [10]Department of Pediatrics, Dell Medical School, The University of Texas at Austin, Austin, TX 78723, USA. [11]These authors contributed equally: Henriette U. Balinda, William J. Kelly. ✉e-mail: brennera@uthscsa.edu

