## [Peer Review File · Nature Communications]

Reviewers' Comments:

Reviewer #1:

Remarks to the Author:

Review

Sacituzumab Govitecan in Patients with Breast Cancer Brain Metastases and Recurrent Glioblastoma

Balinda et al report results of a prospective phase 0 trial investigating the intratumoral concentrations and intracranial activity of SG in patients with brain metastases (BM) of mixed breast cancer subtypes or recurrent glioblastoma (GBM). Patients scheduled for neurosurgical resection of intracranial lesions received a single dose of sacituzumab govitecan (SG) at the standard dose of 10 mg/kg body weight on the day before surgery and treatment was continued postoperatively in order to assess intracranial response in unresected lesions as well as progression-free survival (PFS) and overall survival (OS). SG activity was also evaluated in an intracranial xenograft model, in addition Trop-2 expression, DNA damage (γ -H2AX expression) and intratumoral hypoxia (CAIX expression) was determined in patient tumour tissue.

SG treated animals had a significantly improved OS compared with control animals. An overall number of 25 patients were accrued – 13 patients with breast cancer BM (12 evaluable), 12 patients with GBM. Tissue concentrations of SN-38 were lower compared with serum but exceeded the minimum IC50 for SN-38 in both BC and GBM cell lines. Median OS was 35.2 months in patients with breast cancer BM and 9.5 months in the GBM cohort, respectively. Corresponding numbers regarding response rate were 50% and 57%, respectively. Neither γ -H2AX nor CAIX correlated with % SN-38 tissue/serum ratio.

The authors must be commended for conducting this trial. While meanwhile the intracranial activity of ADCs is well established (e.g., KAMILLA; DEBBRAH; TUXEDO-1), these studies did not evaluate payload tissue concentrations. In addition, data on intracranial activity of SG is limited. Regarding secondary endpoints, interpretation of PFS and OS in the breast cancer cohort is limited by the inclusion of different subtypes; especially the low number of triple-negative breast cancer patients is disappointing, but this does not limit the overall relevance of these data. Authors evaluated preclinical activity of SG in a parallel intracranial xenograft model and found no correlation of CAIX expression with % SN-38 tissue/serum ratio, questioning the role of hypoxia as a mechanisms of SN-38 release in breast metastases. The manuscript is well written and the study design adequate. The limitations of the submitted manuscript (e.g., inclusion of mixed breast cancer subtypes) should be clearly addressed. Overall, the conclusion is supported by the data.

Specific remarks:

1. Introduction: The introduction section is somewhat long and partially out of focus
2. Had all breast cancer BM patients active BM? This is suggested by the fact that RANO-BM criteria were applied for response assessment but seems not to be clearly mentioned in the manuscript.
3. What percentage of BM patients had prior radiotherapy to the resected lesion?
4. What was the interval from radiotherapy to resection?

Thank you for giving me the opportunity to review this manuscript! Kind regards, Rupert Bartsch

Reviewer #2:

Remarks to the Author:

The authors provide a window of opportunity study on SG in breast cancer brain metastases and recurrent glioblastoma with some preclinical and translational data. The rationale is clear and

relevant. No formal sample size calculation was performed and non-SG-treated control groups are not available as no randomization was performed. Still, the data are of interest and will facilitate development of ADCs for intracranial tumors.

The authors should also discuss in more detail the CNS activity of ADCs in general, as well as other TROP-2 targeting ADCs under evaluation for breast cancer brain metastases or glioblastoma, the most comprehensive reference on the topic is missing: PMID: 37085569

WHO 2021 classification there are no IDH-mutant glioblastomas. Yet, here IDH mutant cases were enrolled. Please clarify. Of note, the abbreviation "GBM" is discouraged by the current WHO classification.

Which edition of the RANO criteria were used for radiological assessments?

MRI images of objective responses in breast cancer and glioblastoma cases would be helpful for visualisation.

What is the rationale for the pre-surgical time interval of SG administration?

The abstract seems abbreviated and does not contain the preclinical and translational data.

Reviewer #3:

Remarks to the Author:

The ms reports on a prospective, window-of-opportunity trial (NCT03995706) to examine the intra-tumoral concentrations, as a primary endpoint, and intracranial activity (PFS) and OS, as secondary endpoints, of Sacituzumab Govitecan (SG) in 13 patients with breast cancer brain metastases (BCBM) and 12 patients with recurrent glioblastoma (rGBM). The median molarity achieved in BCBM tumor tissue was approximately 10 times the IC50 measured for BCBM cell lines. The median molarity achieved in rGBM tumor tissue was approximately 60 times the IC50 measured for rGBM cell lines. Thus, success for the primary endpoint was achieved. The median PFS and OS were 8 and 35 months, respectively, for the BCBM cohort and 2 and 9.5 months, respectively, for the rGBM cohort, which was considered promising. The statistical results are described appropriately and adequately and correspond to the statistical design in the study protocol. I have no comments or concerns.

Revision: NCOMMS-24-14698A

REVIEWER #1 COMMENTS

Sacituzumab Govitecan in Patients with Breast Cancer Brain Metastases and Recurrent Glioblastoma

Balinda et al report results of a prospective phase 0 trial investigating the intratumoral concentrations and intracranial activity of SG in patients with brain metastases (BM) of mixed breast cancer subtypes or recurrent glioblastoma (GBM). Patients scheduled for neurosurgical resection of intracranial lesions received a single dose of sacituzumab govitecan (SG) at the standard dose of 10 mg/kg body weight on the day before surgery and treatment was continued post postoperatively in order to assess intracranial response in unresected lesions as well as progression-free survival (PFS) and overall survival (OS). SG activity was also evaluated in an intracranial xenograft model, in addition Trop-2 expression, DNA damage (γ -H2AX expression) and intratumoral hypoxia (CAIX expression) was determined in patient tumour tissue.

SG treated animals had a significantly improved OS compared with control animals. An overall number of 25 patients were accrued – 13 patients with breast cancer BM (12 evaluable), 12 patients with GBM. Tissue concentrations of SN-38 were lower compared with serum but exceeded the minimum IC50 for SN-38 in both BC and GBM cell lines. Median OS was 35.2 months in patients with breast cancer BM and 9.5 months in the GBM cohort, respectively. Corresponding numbers regarding response rate were 50% and 57%, respectively. Neither γ -H2AX nor CAIX correlated with % SN-38 tissue/serum ratio.

The authors must be commended for conducting this trial. While meanwhile the intracranial activity of ADCs is well established (e.g., KAMILLA; DEBBRAH; TUXEDO-1), these studies did not evaluate payload tissue concentrations. In addition, data on intracranial activity of SG is limited. Regarding secondary endpoints, interpretation of PFS and OS in the breast cancer cohort is limited by the inclusion of different subtypes; especially the low number of triple-negative breast cancer patients is disappointing, but this does not limit the overall relevance of these data. Authors evaluated preclinical activity of SG in a parallel intracranial xenograft model and found no correlation of CAIX expression with % SN-38 tissue/serum ratio, questioning the role of hypoxia as a mechanisms of SN-38 release in breast metastases. The manuscript is well written and the study design adequate. The limitations of the submitted manuscript (e.g., inclusion of mixed breast cancer subtypes) should be clearly addressed. Overall, the conclusion is supported by the data.

Specific remarks:

1. Introduction: The introduction section is somewhat long and partially out of focus

The abstract has been edited for a more focused and relevant context of the development of SG.

2. Had all breast cancer BM patients active BM? This is suggested by the fact that RANO-BM criteria were applied for response assessment but seems not to be clearly mentioned in the manuscript.

Yes, of the 8 BCBM patients included in the radiographic response rate analysis (#1, 5, 8, 12, 14, 20, 21, 26), all had active brain metastasis at time of study enrollment. This information is now included in the supplemental section of demographics.

3. What percentage of BM patients had prior radiotherapy to the resected lesion?

2 BCBM patients (#1, 21) had had prior intracranial radiation including whole brain radiation. These were the only two BCBM with prior radiation and made up 15% of that cohort.

4. What was the interval from radiotherapy to resection?

For both of these patients, the interval between last radiation and surgical resection on study was 273 and 230 days, respectively. Information is now included in the supplemental section of demographics.

5. The limitations of the submitted manuscript (e.g., inclusion of mixed breast cancer subtypes) should be clearly addressed.

As the reviewer rightly points out, interpretation of PFS and OS is limited by the inclusion of different subtypes. As a phase 0, a trade off prioritizing feasibility and generalizability at the expense of specificity was felt necessary during the design of inclusion/exclusion criteria. This study's findings should be confirmed with a dedicated phase 3 study enrolling a number of patients of each breast subtype with adequate statistical power to address these concerns. This consideration has been added to the discussion section of the paper.

Thank you for giving me the opportunity to review this manuscript! Kind regards, Rupert Bartsch

REVIEWER #2 COMMENTS

The authors provide a window of opportunity study on SG in breast cancer brain metastases and recurrent glioblastoma with some preclinical and translational data. The rationale is clear and relevant. No formal sample size calculation was performed and non-SG-treated control groups are not available as no randomization was performed. Still, the data are of interest and will facilitate development of ADCs for intracranial tumors.

1. The authors should also discuss in more detail the CNS activity of ADCs in general, as well as other TROP-2 targeting ADCs under evaluation for breast cancer brain metastases or glioblastoma, the most comprehensive reference on the topic is missing: PMID: 37085569

The authors appreciate such a comprehensive reference, this review has been included in the background as well as data pertaining to KAMILLA, DEBBRAH and TUXEDO-1.

2. WHO 2021 classification there are no IDH-mutant glioblastomas. Yet, here IDH mutant cases were enrolled. Please clarify. Of note, the abbreviation "GBM" is discouraged by the current WHO classification.

Yes, the study design, initiation and name unfortunately predated these changes. Three patients had IDH-mutant tumors and one patient had IDH-wild-type molecular glioblastoma. This was declared in the supplemental, but we have now moved this to the main body for better transparency. In order to avoid confusing the reader we have left cohort B as "rGBM". However, if the reviewers feel strongly, we would also be happy to rename as "recurrent glioma" throughout.

3. Which edition of the RANO criteria were used for radiological assessments?

Rano 1.0 was used for glioma. Clarification of this and the relevant paper has been referenced in the Methods section.

4. MRI images of objective responses in breast cancer and glioblastoma cases would be helpful for visualization.

Yes, these have been added as Supplementary Figures in the Results: Clinical Efficacy section.

5. What is the rationale for the pre-surgical time interval of SG administration?

A serum half-life of 11-14 hours was observed in the phase ⅓ study (Ocean AJ et al, Cancer 2017). Therefore, a pre-surgical interval of roughly 24 hours was deemed appropriate to measure intracranial penetration while accounting for operating room logistics. Mention of this rationale has been added to the methods section.

6. The abstract seems abbreviated and does not contain the preclinical and translational data.

Our apologies. The abstract was originally submitted to Nature Cancer which limits article abstracts to 150 words. The relevant preclinical and translational data has been added back to the abstract.

REVIEWER #3 COMMENTS

The ms reports on a prospective, window-of-opportunity trial (NCT03995706) to examine the intra-tumoral concentrations, as a primary endpoint, and intracranial activity (PFS) and OS, as secondary endpoints, of Sacituzumab Govitecan (SG) in 13 patients with breast cancer brain metastases (BCBM) and 12 patients with recurrent glioblastoma (rGBM). The median molarity achieved in BCBM tumor tissue was approximately 10 times the IC50 measured for BCBM cell lines. The median molarity achieved in rGBM tumor tissue was approximately 60 times the IC50 measured for rGBM cell lines. Thus, success for the primary endpoint was achieved. The median PFS and OS were 8 and 35 months, respectively, for the BCBM cohort and 2 and 9.5 months, respectively, for the rGBM cohort, which was considered promising. The statistical results are described appropriately and adequately and correspond to the statistical design in the study protocol. I have no comments or concerns.

Minor Corrections: An error was noted during review where DCR had been incorrectly reported as ORR for GBM. We have corrected this to read ORR 28% (28% SD, 28% PR, 0% CR) including in the abstract. The iORR for the breast cancer cohort remains unchanged. Information has also been included in Supplementary Demographics on one rGBM patient who had radiation necrosis on pathology but elected to continue on study. A patient (#11) who had been given a study ID but never treated on study was inadvertently included on the demographics table (but not other analysis). This has been corrected. Mention is also made of 1 patient (#15) who had a non-canonical G105G IDH mutation which could not be verified with 2 separate comprehensive molecular profiling tests.

Reviewers' Comments:

Reviewer #1:

Remarks to the Author:

Thank you, all concerns were addressed and I have no further comments.

Kind regards,

Rupert Bartsch

Reviewer #2:

Remarks to the Author:

The authors have addressed my concerns adequately.

Revision: NCOMMS-24-14698A

REVIEWER #1 COMMENTS

Sacituzumab Govitecan in Patients with Breast Cancer Brain Metastases and Recurrent Glioblastoma

Balinda et al report results of a prospective phase 0 trial investigating the intratumoral concentrations and intracranial activity of SG in patients with brain metastases (BM) of mixed breast cancer subtypes or recurrent glioblastoma (GBM). Patients scheduled for neurosurgical resection of intracranial lesions received a single dose of sacituzumab govitecan (SG) at the standard dose of 10 mg/kg body weight on the day before surgery and treatment was continued post postoperatively in order to assess intracranial response in unresected lesions as well as progression-free survival (PFS) and overall survival (OS). SG activity was also evaluated in an intracranial xenograft model, in addition Trop-2 expression, DNA damage (γ -H2AX expression) and intratumoral hypoxia (CAIX expression) was determined in patient tumour tissue.

SG treated animals had a significantly improved OS compared with control animals. An overall number of 25 patients were accrued – 13 patients with breast cancer BM (12 evaluable), 12 patients with GBM. Tissue concentrations of SN-38 were lower compared with serum but exceeded the minimum IC50 for SN-38 in both BC and GBM cell lines. Median OS was 35.2 months in patients with breast cancer BM and 9.5 months in the GBM cohort, respectively. Corresponding numbers regarding response rate were 50% and 57%, respectively. Neither γ -H2AX nor CAIX correlated with % SN-38 tissue/serum ratio.

The authors must be commended for conducting this trial. While meanwhile the intracranial activity of ADCs is well established (e.g., KAMILLA; DEBBRAH; TUXEDO-1), these studies did not evaluate payload tissue concentrations. In addition, data on intracranial activity of SG is limited. Regarding secondary endpoints, interpretation of PFS and OS in the breast cancer cohort is limited by the inclusion of different subtypes; especially the low number of triple-negative breast cancer patients is disappointing, but this does not limit the overall relevance of these data. Authors evaluated preclinical activity of SG in a parallel intracranial xenograft model and found no correlation of CAIX expression with % SN-38 tissue/serum ratio, questioning the role of hypoxia as a mechanisms of SN-38 release in breast metastases. The manuscript is well written and the study design adequate. The limitations of the submitted manuscript (e.g., inclusion of mixed breast cancer subtypes) should be clearly addressed. Overall, the conclusion is supported by the data.

Specific remarks:

1. Introduction: The introduction section is somewhat long and partially out of focus

The abstract has been edited for a more focused and relevant context of the development of SG.

2. Had all breast cancer BM patients active BM? This is suggested by the fact that RANO-BM criteria were applied for response assessment but seems not to be clearly mentioned in the manuscript.

Yes, of the 8 BCBM patients included in the radiographic response rate analysis (#1, 5, 8, 12, 14, 20, 21, 26), all had active brain metastasis at time of study enrollment. This information is now included in the supplemental section of demographics.

3. What percentage of BM patients had prior radiotherapy to the resected lesion?

2 BCBM patients (#1, 21) had had prior intracranial radiation including whole brain radiation. These were the only two BCBM with prior radiation and made up 15% of that cohort.

4. What was the interval from radiotherapy to resection?

For both of these patients, the interval between last radiation and surgical resection on study was 273 and 230 days, respectively. Information is now included in the supplemental section of demographics.

5. The limitations of the submitted manuscript (e.g., inclusion of mixed breast cancer subtypes) should be clearly addressed.

As the reviewer rightly points out, interpretation of PFS and OS is limited by the inclusion of different subtypes. As a phase 0, a trade off prioritizing feasibility and generalizability at the expense of specificity was felt necessary during the design of inclusion/exclusion criteria. This study's findings should be confirmed with a dedicated phase 3 study enrolling a number of patients of each breast subtype with adequate statistical power to address these concerns. This consideration has been added to the discussion section of the paper.

Thank you for giving me the opportunity to review this manuscript! Kind regards, Rupert Bartsch

REVIEWER #2 COMMENTS

The authors provide a window of opportunity study on SG in breast cancer brain metastases and recurrent glioblastoma with some preclinical and translational data. The rationale is clear and relevant. No formal sample size calculation was performed and non-SG-treated control groups are not available as no randomization was performed. Still, the data are of interest and will facilitate development of ADCs for intracranial tumors.

1. The authors should also discuss in more detail the CNS activity of ADCs in general, as well as other TROP-2 targeting ADCs under evaluation for breast cancer brain metastases or glioblastoma, the most comprehensive reference on the topic is missing: PMID: 37085569

The authors appreciate such a comprehensive reference, this review has been included in the background as well as data pertaining to KAMILLA, DEBBRAH and TUXEDO-1.

2. WHO 2021 classification there are no IDH-mutant glioblastomas. Yet, here IDH mutant cases were enrolled. Please clarify. Of note, the abbreviation "GBM" is discouraged by the current WHO classification.

Yes, the study design, initiation and name unfortunately predated these changes. Three patients had IDH-mutant tumors and one patient had IDH-wild-type molecular glioblastoma. This was declared in the supplemental, but we have now moved this to the main body for better transparency. In order to avoid confusing the reader we have left cohort B as "rGBM". However, if the reviewers feel strongly, we would also be happy to rename as "recurrent glioma" throughout.

3. Which edition of the RANO criteria were used for radiological assessments?

Rano 1.0 was used for glioma. Clarification of this and the relevant paper has been referenced in the Methods section.

4. MRI images of objective responses in breast cancer and glioblastoma cases would be helpful for visualization.

Yes, these have been added as Supplementary Figures in the Results: Clinical Efficacy section.

5. What is the rationale for the pre-surgical time interval of SG administration?

A serum half-life of 11-14 hours was observed in the phase ⅓ study (Ocean AJ et al, Cancer 2017). Therefore, a pre-surgical interval of roughly 24 hours was deemed appropriate to measure intracranial penetration while accounting for operating room logistics. Mention of this rationale has been added to the methods section.

6. The abstract seems abbreviated and does not contain the preclinical and translational data.

Our apologies. The abstract was originally submitted to Nature Cancer which limits article abstracts to 150 words. The relevant preclinical and translational data has been added back to the abstract.

REVIEWER #3 COMMENTS

The ms reports on a prospective, window-of-opportunity trial (NCT03995706) to examine the intra-tumoral concentrations, as a primary endpoint, and intracranial activity (PFS) and OS, as secondary endpoints, of Sacituzumab Govitecan (SG) in 13 patients with breast cancer brain metastases (BCBM) and 12 patients with recurrent glioblastoma (rGBM). The median molarity achieved in BCBM tumor tissue was approximately 10 times the IC50 measured for BCBM cell lines. The median molarity achieved in rGBM tumor tissue was approximately 60 times the IC50 measured for rGBM cell lines. Thus, success for the primary endpoint was achieved. The median PFS and OS were 8 and 35 months, respectively, for the BCBM cohort and 2 and 9.5 months, respectively, for the rGBM cohort, which was considered promising. The statistical results are described appropriately and adequately and correspond to the statistical design in the study protocol. I have no comments or concerns.

Minor Corrections: An error was noted during review where DCR had been incorrectly reported as ORR for GBM. We have corrected this to read ORR 28% (28% SD, 28% PR, 0% CR) including in the abstract. The iORR for the breast cancer cohort remains unchanged. Information has also been included in Supplementary Demographics on one rGBM patient who had radiation necrosis on pathology but elected to continue on study. A patient (#11) who had been given a study ID but never treated on study was inadvertently included on the demographics table (but not other analysis). This has been corrected. Mention is also made of 1 patient (#15) who had a non-canonical G105G IDH mutation which could not be verified with 2 separate comprehensive molecular profiling tests.